# Beyond Mosquitoes: A Review of Pediatric Vector-Borne Diseases Excluding Malaria and Arboviral Infections

**DOI:** 10.3390/pathogens14060553

**Published:** 2025-06-02

**Authors:** Giulia Carbone, Amina De Bona, Dragos Septelici, Alessandro Cipri, Andrea Nobilio, Susanna Esposito

**Affiliations:** Pediatric Clinic, Department of Medicine and Surgery, University of Parma, 43126 Parma, Italy; giulia.carbone@unipr.it (G.C.); amina.debona@unipr.it (A.D.B.); dragos.septelici@unipr.it (D.S.); alessandro.cipri@unipr.it (A.C.); andrea.nobilio@unipr.it (A.N.)

**Keywords:** vector-borne diseases, arthropod bites, tick-borne diseases, sandfly-transmitted diseases, bedbug infestations, flea-borne infections, climate change, emerging zoonotic infections

## Abstract

Vector-borne diseases (VBDs) significantly impact global child health, with mosquito-transmitted infections like malaria and arboviruses accounting for a substantial portion of this burden. However, other arthropod-borne diseases—transmitted by vectors such as ticks, fleas, sand flies, lice, and triatomine bugs—also pose serious health risks to children worldwide. This review specifically excludes mosquito-borne diseases to concentrate on these less-discussed yet clinically important pediatric VBDs. We examine their clinical manifestations, diagnostic challenges, and treatment options, highlighting the unique vulnerabilities of children, including immature immune systems, behavioral factors, and communication barriers that can delay diagnosis. Additionally, we explore how environmental and anthropogenic factors, such as climate change and urbanization, are expanding the geographic range of these vectors, leading to the emergence of diseases like Lyme disease and leishmaniasis in new regions. By focusing on non-mosquito VBDs, this review aims to raise awareness and inform healthcare providers and public health practitioners about the comprehensive landscape of pediatric vector-borne diseases.

## 1. Introduction

Arthropod-borne diseases represent a significant global health challenge, particularly among children, who are disproportionately affected by these infections. Several arthropods act as vectors for infectious diseases, including ticks, fleas, sand flies, and bedbugs [1,2]. These vectors can transmit a variety of pathogens, leading to conditions that range from mild febrile illnesses to severe, life-threatening diseases [3,4]. Children are particularly vulnerable due to several factors, including their increased exposure to outdoor environments and close contact with animals [5]. Additionally, the immaturity of their immune systems renders them less capable of mounting effective defenses against infections [6], while behaviors such as scratching further elevate the risk of secondary bacterial infections [7]. Furthermore, children often struggle to communicate early, nonspecific symptoms—such as fever, fatigue, or rash—leading to delayed diagnosis and treatment, which can exacerbate disease severity [8].

The burden of vector-borne diseases is further intensified by environmental and anthropogenic factors, including climate change, deforestation, urbanization, and increased human–wildlife interactions [9,10]. Rising global temperatures have expanded the habitats of various arthropod vectors, facilitating their spread into previously unaffected regions [11]. For instance, Lyme disease, once restricted to specific areas of North America and Europe, has now been reported in new geographic regions due to the increasing distribution of ticks [12]. Similarly, leishmaniasis, historically endemic to tropical and subtropical areas, is now emerging in temperate climates as the expansion of sand fly populations accelerates in response to global warming [13,14].

This narrative review aims to describe pediatric VBDs transmitted by non-mosquito arthropods (specifically ticks, fleas, sand flies, and bedbugs). Our focus is on their clinical manifestations in children, diagnostic challenges, treatment options, and preventive strategies pertinent to pediatric populations. Notably, we intentionally excluded mosquito-borne diseases, such as malaria and arboviral infections (e.g., dengue, Zika, chikungunya), as these have been extensively reviewed elsewhere and represent a distinct category of VBDs. While formal guidelines, like PRISMA, are tailored for systematic reviews, narrative reviews lack standardized reporting protocols. Nevertheless, we endeavored to maintain methodological transparency and minimize bias by adhering to a structured approach. We conducted a comprehensive electronic search of the PubMed/MEDLINE and Cochrane Central databases, covering literature published from 1 January 2010 to 31 December 2024. The search utilized combinations of the following keywords: “vector”, “insects”, “tick”, “flea”, “sand fly”, “bedbug”, “children”, “adolescent”, “pediatric”, and “pediatric”. We applied filters to include only English-language articles and excluded studies focusing on mosquito-borne diseases. Additionally, we manually screened the references of relevant articles to identify further pertinent studies. Inclusion criteria encompassed peer-reviewed articles that provided clinical data on pediatric populations affected by non-mosquito VBDs, including case reports, case series, observational studies, and reviews. Exclusion criteria were studies centered on adult populations, non-human subjects, or those lacking specific data on pediatric clinical manifestations. By delineating our methodology and scope, we aim to offer a focused and informative overview of pediatric VBDs beyond those transmitted by mosquitoes.

## 2. Tick Bites

Ticks are hematophagous arthropods that pose a significant health risk to children due to their ability to transmit infectious diseases [15]. Most medically relevant tick species belong to the Ixodes genus, though the specific species vary by geographical region [1]. Young children are particularly susceptible to tick-borne infections due to their curiosity, frequent outdoor activities, and close contact with animals. Diseases such as Lyme disease and tick-borne encephalitis (TBE) can have severe and long-lasting consequences if left untreated [16]. Tick populations are increasing and expanding into new regions due to factors such as climate change, habitat fragmentation, and rising human–wildlife interactions, making human and animal exposure more frequent [1].

Tick-borne infections are often under-reported, partly due to missed diagnoses or cross-reactions with other rickettsial infections of uncertain pathogenicity [16]. Another challenge in detection is that tick bites are painless, making them difficult to notice [17]. Males are more frequently affected [17], and risk factors include outdoor activities, such as hiking, camping, and gardening, as well as exposure to domestic or wild animals [16,17]. Urban exposure is also an emerging concern, as ticks are increasingly found in parks and suburban gardens. In the Northern Hemisphere, most tick-borne disease cases occur between April and October, aligning with increased tick activity and human outdoor exposure during warmer months [17]. However, tick activity can persist year-round in some regions, particularly in milder climates. Tick-borne infections are endemic in various parts of the world, with high Lyme disease incidence in countries such as Sweden, Norway, Germany, France, and the United States, while TBE is endemic in large regions of Russia, Central Europe, and parts of Asia [1,15,16,17].

Ticks, which are arachnids, serve as vectors for numerous bacterial, viral, and protozoan pathogens. Once attached to a host, ticks feed on blood, transmitting pathogens through infected saliva [1]. The size and appearance of the bite depend on the tick species and the duration of attachment, but they are often small and inconspicuous [15]. Bites are more common on the ankles, behind the knees, and the groin in adults; whereas, in children, they are frequently found on the head, neck, and scalp [17]. The bite site may exhibit redness, swelling, and itching [17]. The likelihood of contracting a tick-borne infection depends on several factors, including tick species, the duration of attachment, and the pathogen load within the tick’s saliva [17]. Table 1 summarizes the risk factors of tick-borne disease.

Prompt and proper tick removal is crucial to reducing the risk of infection. The recommended technique involves using fine-tipped tweezers to grasp the tick as close to the skin as possible, pulling steadily and firmly upward without twisting to avoid crushing the tick and releasing pathogens. The bite area should then be washed with antibacterial soap and disinfected with alcohol or a chlorhexidine-based antiseptic [18]. According to NICE guidelines, a history of a tick bite alone should not prompt a Lyme disease diagnosis, and asymptomatic patients do not require treatment. In non-endemic areas, more than 85% of ticks are uninfected, meaning most bites are harmless [18]. Symptoms of tick-borne infections in children can be nonspecific, often resembling viral illnesses. Common signs include erythema migrans (a red expanding rash), fever, headache, fatigue, nausea, vomiting, joint and muscle pain, and swollen lymph nodes [16]. Different tick-borne infections present with unique symptoms, making diagnosis challenging.

Ticks transmit a range of pathogens, including spirochetes (*Borrelia* spp., causing Lyme disease), rickettsiae (*Rickettsia*, *Ehrlichia*, *Anaplasma*), bacteria (*Francisella tularensis*, causing tularemia), protozoa (*Babesia* spp., causing babesiosis), and viruses (tick-borne encephalitis virus, Colorado tick fever virus) [1,19]. Lyme borreliosis is the most common tick-borne disease in the Northern Hemisphere, caused by *Borrelia burgdorferi* and transmitted through Ixodes tick bites [20,21,22]. Children are at higher risk due to greater outdoor exposure. The disease progresses in stages, beginning with early localized infection, which typically presents as erythema migrans (EM), appearing 7–14 days after the bite. The rash may expand and develop a bull’s eye appearance, and other symptoms such as fever, malaise, headache, and lymphadenopathy may occur [23,24,25,26,27]. If left untreated, within 3–5 weeks, the infection can spread to the nervous system, heart, and joints, leading to complications, such as facial palsy, meningitis, encephalopathy, Lyme carditis (heart block, syncope, dizziness), and arthritis, most commonly affecting the knees [26,27]. Late Lyme disease (LLD) develops weeks to months later if untreated, causing persistent arthritis, musculoskeletal pain, fatigue, and cognitive issues, often referred to as post-treatment Lyme disease syndrome (PTLDS) [25].

Lyme disease is primarily diagnosed clinically, with laboratory testing used for confirmation. The diagnostic approach includes an initial enzyme immunoassay (EIA) or immunofluorescence assay (IFA), followed by a confirmatory Western blot [19,27]. In early disease, antibodies may be undetectable, leading to false negatives, particularly in children under five years old [27]. In cases presenting with neurological or cardiac symptoms, cerebrospinal fluid analysis or ECG may aid in the diagnosis [27]. Treatment aims to reduce symptom duration and prevent dissemination. Early localized disease is treated with oral antibiotics, such as doxycycline (for children over eight years old) or amoxicillin and cefuroxime axetil (for children under eight years old) [18,27]. For disseminated disease, intravenous antibiotics, such as ceftriaxone or cefotaxime, are required [18,27]. European trials have demonstrated that azithromycin (500 mg on day one, followed by 250 mg for four additional days) and doxycycline (100 mg twice daily for 14 days) yield similar results in treating early Lyme disease [18,27]. Treatment duration generally ranges from 14 to 21 days, depending on disease severity.

Table 2 describes the most common clinical manifestations and recommended treatment of major tick-borne diseases.

Preventing tick-borne diseases requires multiple protective measures. The use of tick repellents, such as DEET (20–30%), on exposed skin and permethrin-treated clothing is recommended [18]. Wearing light-colored, long-sleeved clothing allows for easier detection of ticks, and thorough tick checks after outdoor exposure are essential. Environmental modifications, such as keeping grass short and removing brush near frequented areas, help minimize tick habitats. Household pets should be checked for ticks and treated with veterinary-approved tick prevention products, as they can bring ticks into the home [19]. Awareness of symptoms and early medical intervention in endemic areas are key to reducing complications. Tick-borne diseases continue to pose a growing public health concern, particularly among children. Early recognition, proper tick removal, and timely treatment are critical in preventing severe complications. While most tick bites are harmless, efforts in education, prevention, and surveillance remain essential in mitigating the risk of these infections.

## 3. Sandfly Bite

*Phlebotominae*, commonly known as sandflies, are noiseless hematophagous arthropods approximately 2–3 mm in size, varying in color from white to black. In the Mediterranean region, sandflies are mainly active during the wet summer months [28]. Most *Phlebotominae* bite outdoors from dusk till dawn, though some species also bite indoors during the day. Due to their limited vertical jumping ability, they are less likely to bite individuals residing on the upper floors of buildings [28]. In the United States, sandflies are found in several regions of the South [29].

A sandfly bite can transmit several infectious diseases, the most significant being leishmaniasis, a neglected tropical disease caused by more than 20 species of obligate intracellular protozoa belonging to the genus *Leishmania* [28,29]. The primary pathogenic species include L. *donovani*, *L. infantum*, *L. chagasi*, *L. tropica*, *L. major*, *L. aethiopica*, *L. braziliensis*, *L. guyanensis*, and *L. panamensis* [28]. Transmission occurs when a female sandfly feeds on a human host and injects promastigotes into the dermis. The parasites are then phagocytized by macrophages, where they transform into amastigotes and parasitize different cell populations of the human host [28,29]. Sandfly distribution and density are closely associated with temperature and humidity, making leishmaniasis transmission susceptible to climate change [30,31].

Currently, no vaccine is available for human leishmaniasis, and no standardized measures exist to control sandfly populations [31]. The cutaneous form (CL) is the most common, while visceral leishmaniasis (VL) has the highest mortality rates, especially if untreated. Mucosal leishmaniasis (ML), though rarely fatal, causes significant disfigurement and decreased quality of life in affected individuals [32,33]. Globally, leishmaniasis affects approximately 4.7 million people, with an annual incidence of 700,000 to 1,000,000 new cases [30,31,32]. According to the WHO, 71 countries are endemic for both cutaneous and visceral leishmaniasis. In 2023, Brazil, Peru, Algeria, Syria, and Pakistan had the highest reported CL cases (≥5000); whereas, Brazil, Sudan, and Ethiopia had the highest VL cases (≥1000) [31].

Leishmaniasis significantly impacts children, with 30.2–46.6% of cases occurring in pediatric populations. In the Americas, more than 30% of CL and ML cases occur in individuals under 20 years of age, while 13% occur in children under 10 years old [33]. Regarding VL, 43% of cases occur in individuals under 19 years, and 35% in children under 10 years. It is the second leading parasitic cause of death in children under 5 years old, after malaria, and the third leading cause in children aged 5–14 years, accounting for an estimated 3338 pediatric deaths and 382,634 disability-adjusted life years (DALYs) in 2019 [33]. Younger children are particularly vulnerable due to immature cell-mediated immunity and a lack of previous exposure to *Leishmania* antigens [34].

The best way to prevent leishmaniasis is to protect exposed body surfaces from sandfly bites. Insect repellent application is recommended, particularly for children engaging in recreational activities from dusk till dawn [34]. The incidence of cutaneous leishmaniasis is rising globally due to deforestation, urbanization, and human migration, which alter host immunity and vector habitats [35,36,37,38]. CL can present in varied clinical forms, from single localized lesions to widespread dissemination. Lesions usually appear one week after a sandfly bite at exposed sites, such as the face, head, arms, and legs [33]. The lesion progresses from a papule to a nodule, ulcerates within 5–7 months, and heals with scarring over 8–12 months [33]. Some infections resolve spontaneously depending on the *Leishmania* species [33]. In pediatric patients, CL lesions are more frequently found on the head and neck compared to adults [34].

Complications include relapsing leishmaniasis, where new lesions form at the periphery of healed scars, typically caused by *L. tropica*, and disseminated CL, which requires systemic treatment [34,38]. Mucosal leishmaniasis (ML) is primarily caused by *L. braziliensis*, *L. guyanensis*, and *L. panamensis* [35,36], with rare cases involving *L. aethiopica*. ML often develops months to years after cutaneous lesions, initially presenting as nasal congestion, epistaxis, hyposmia, or nasal obstruction [37]. If untreated, it can progress to severe mucosal destruction, particularly affecting the nasal septum and causing facial disfigurement [38].

Visceral leishmaniasis (VL), or kala-azar, is primarily caused by *L. donovani* in India and Eastern Africa; whereas, *L. infantum* predominates in the Mediterranean, Middle East, and Latin America [29,35,36,37]. In Southern Europe, VL is endemic, with recent outbreaks reported near urban areas in Madrid (Spain) and Bologna (Italy) [39,40]. The Tyrrhenian coast of Italy is one of the most endemic areas, with seroprevalence rates of up to 40% in dogs, which serve as the primary reservoir [40]. Although dogs are the main reservoir, evidence suggests that cats, rodents, and other wild mammals may also harbor *Leishmania* [40]. In Europe, a canine leishmaniasis vaccine is available but requires pre-vaccination serological screening [31].

VL has an incubation period of weeks to years, though symptoms usually develop within 2–6 months. It presents with fever, fatigue, weight loss, hepatosplenomegaly, lymphadenopathy, and pancytopenia due to parasite accumulation in the bone marrow, spleen, and liver [33,34,38]. Common lab findings include normocytic anemia, thrombocytopenia, hypergammaglobulinemia, hypoalbuminemia, and transaminitis [34,38]. Severe VL may result in spontaneous bleeding, disseminated intravascular coagulation (DIC), liver failure, and secondary hemophagocytic lymphohistiocytosis (HLH) [33,34,38].

Diagnosis of cutaneous and mucosal leishmaniasis requires lesion scraping, biopsy, or aspirate, followed by Giemsa staining, PCR, or culture testing for *Leishmania* [34]. VL diagnosis involves splenic or bone marrow aspiration, serology, and immunochromatographic tests for rk39 antigen [34]. Treatment of uncomplicated CL in immunocompetent patients can follow a “watch-and-wait” approach [34]. For extensive lesions, lesions on functionally or cosmetically sensitive sites, or ML-risk species, systemic therapy is preferred. Intravenous liposomal amphotericin B is the standard treatment in the U.S. for both CL and ML, while oral miltefosine is FDA-approved for children over 12 years and 30 kg [33,41]. VL treatment typically requires intravenous liposomal amphotericin B (3 mg/kg/day for 3 weeks in immunocompetent patients, 4 mg/kg/day for 5 weeks in immunocompromised individuals) [34].

Finally, sandflies can also transmit sandfly fever, a self-limiting arboviral illness caused by sandfly fever virus (SFV), an RNA phlebovirus [42]. The four main serotypes include Sandfly Sicilian virus, Sandfly Cyprus virus, Sandfly Naples virus, and Toscana virus (TOSV), the latter of which is associated with viral meningitis in Southern Europe [41]. Clinically, sandfly fever presents with fever, rash, myalgia, nausea, leukopenia, and thrombocytopenia and is diagnosed through serology. Treatment is supportive, including fluid therapy, bed rest, and analgesia [43].

## 4. *Cimicidae* Bites

Bedbugs and related hematophagous arthropods belong to the infraorder Cimicomorpha, within the suborder *Heteroptera* of the order *Hemiptera* [41,43,44]. The family *Cimicidae*, which includes bedbugs (*Cimex* spp.), is part of the superfamily *Cimicoidea*, alongside other medically significant families, such as *Polyctenidae* and *Reduviidae* (specifically the subfamily *Triatominae*). Together with the family *Reduviidae*, they play a significant role in the medical field. While other insects of this order indirectly affect humans through agriculture, such as aphids, mealybugs, aleurodids, and psyllids, the primary species responsible for human infestations are bedbugs (*Cimex lectularius* and *Cimex hemipterus*) and triatomines (*Triatoma infestans*, *Triatoma sordida*, and *Triatoma rubrofasciata*) [41,43,44]. These insects cause cutaneous and systemic allergic reactions and are potential vectors for various pathogens, including *Trypanosoma cruzi*, the causative agent of Chagas disease, *Coxiella burnetii*, *Wolbachia* spp., arboviruses, and others [41,43,44].

Bedbugs measure 4 to 7 mm in length and have a flat, oval, reddish-brown, and wingless body [45]. They reproduce through traumatic insemination, in which females lay approximately five eggs daily, producing 200–500 eggs throughout their lifespan. Eggs hatch within 4–12 days, and nymphs pass through five developmental stages before reaching adulthood. Bedbugs can survive for 6–12 months, even without feeding, enduring up to 80–140 days without a blood meal. These evolutionary traits enable their rapid reproduction and infestation expansion, with a small number of bedbugs capable of establishing a significant colony within one month [45].

Bedbugs are typically nocturnal, feeding on exposed body areas and locating their hosts by detecting carbon dioxide, body heat, and pheromonal cues. They are cosmopolitan insects, with infestations reported across all continents [45]. Their spread occurs through active dispersal, allowing room-to-room transmission via ventilation ducts, and passive dispersal, where they are transported over long distances via clothing, luggage, and furniture. Overcrowding and poor living conditions increase the risk of infestation [45].

Bedbug bites are painless due to anesthetic compounds in their saliva, with symptoms manifesting within minutes to several days. The saliva contains anticoagulants (such as a factor-X inhibitor), vasodilators (e.g., nitric oxide), and proteolytic enzymes, facilitating blood extraction [43,44]. The resulting skin lesion is typically an itchy, erythematous maculopapule, varying from 5 mm to 2 cm in diameter, depending on individual sensitivity. Lesions are concentrated in exposed body areas, often appearing in linear or curved patterns, and persist for one to two weeks. The number of lesions depends on the severity of the infestation, ranging from a few isolated bites to multiple lesions. Most cases resolve spontaneously within 2–6 weeks, but some may develop hyperpigmentation, secondary infections, or cellulitis [43,44].

Although bedbugs are not established vectors of human pathogens, studies continue to investigate their potential role in disease transmission, given their similarities to other hematophagous arthropods [45]. With climate change and global migration, new vector–pathogen–host interactions could emerge [45]. While HIV and HBV have been detected in bedbugs after feeding on infected individuals, no evidence supports their multiplication or excretion in feces, making transmission unlikely. However, *Coxiella burnetii*, *Trypanosoma cruzi*, *Wolbachia* spp., and arboviruses remain under investigation [45].

*Coxiella burnetii* is the causative agent of Q fever, primarily transmitted through aerosolized bacterial spores or the consumption of unpasteurized dairy products [46,47]. Research has shown that bedbugs can become infected with *C. burnetii* after feeding on infected animals. The bacterium persists throughout all developmental stages of the bedbug, maintaining pathogenicity for up to 250 days, and is excreted in feces [46,47]. However, direct transmission to humans still requires further study.

*Wolbachia* spp. are obligate intracellular bacterial symbionts that infect various invertebrate species, including bedbugs, and manipulate host reproduction. These bacteria selectively eliminate male specimens, reducing fertility, inducing sex change, or even causing death [48,49]. Studies show that *Wolbachia* infects over 60% of insect species, with its DNA detectable in a majority of arthropods [48,49]. Current research focuses on exploiting *Wolbachia* as a biocontrol strategy to manage bedbug infestations, similar to ongoing studies in *Aedes*, *Anopheles*, and *Culex* mosquito species [48,49].

*Trypanosoma cruzi*, the causative agent of Chagas disease, is a hemoflagellate protozoan transmitted primarily by *Triatominae* bugs, though evidence suggests bedbugs may also serve as vectors [50,51,52]. A pivotal 1982 study by Jörg ME, later expanded in 1992, demonstrated that bedbugs (*C. lectularius*) could acquire *T. cruzi* after feeding on infected rodents. The protozoan persisted within the bedbugs for up to 320 days, and transmission to healthy rats occurred in 96.6% of cases. A 1970s outbreak in Buenos Aires further suggested bedbug involvement in human transmission, with a child contracting Chagas disease after sharing a room with an infected relative during an infestation. A more recent study by Salazar et al. found that bedbugs acquired *T. cruzi* and transmitted it to 40% of exposed mice through fecal contamination [50,51,52].

Chagas disease (CD), or American trypanosomiasis, is the third most prevalent parasitic disease worldwide. Initially endemic to Latin America, it has spread to North America, Europe, and Australia due to migration [53]. Among Latin American migrants in Europe, the estimated *T. cruzi* seroprevalence is 1.8–2.8%, with the highest rates in Spain (2.3–3.8%), Belgium (1.6–2.1%), and Italy (1.6–2%) [54]. Pediatric cases outside endemic regions remain under-reported, but a study of 55,367 migrants estimated a 2.6% prevalence in adults and 0.1% in children under 14 years [55,56]. While vector-mediated transmission has not been reported in Europe, the predominant route of infection in non-endemic countries is vertical transmission from mother to child (4.7%), which increases risks of preterm birth, low birth weight, and perinatal complications [57,58,59,60,61,62,63,64].

The life cycle of *T. cruzi* involves two hosts and multiple developmental stages. The protozoan enters the mammalian host through an arthropod bite wound, where trypomastigotes penetrate host cells, differentiate into amastigotes, and multiply [53,54,55]. After cell lysis, they transform back into trypomastigotes, which disseminate through the bloodstream and infect new cells, initiating acute infection [53,54,55].

Clinically, CD has acute and chronic phases [65,66,67,68,69]. The acute phase, observed in 1–5% of patients, presents with fever, a chagoma at the inoculation site, unilateral palpebral edema (Romaña sign), lymphadenopathy, hepatosplenomegaly, myocarditis, pericardial effusion, and meningoencephalitis, with a 0.2–0.5% mortality risk [65,66,67,68,69]. If untreated, the disease progresses to chronic infection, often remaining asymptomatic. However, 14–45% of patients develop cardiac complications, including arrhythmias, cardiomyopathy, apical aneurysms, heart failure, and thromboembolic events [70,71,72,73,74,75]. Gastrointestinal involvement, affecting 10–21% of patients, causes achalasia, megaesophagus, and megacolon, leading to dysphagia, reflux, and weight loss [70,71,72,73,74,75].

Diagnosis of acute CD is based on microscopic detection of *T. cruzi* in blood smears or PCR; whereas, chronic CD relies on serological IgG testing via ELISA, indirect fluorescent assays, or hemagglutination tests [76,77,78,79,80,81]. Treatment includes Benznidazole and Nifurtimox (Table 3), with early intervention crucial due to reduced efficacy over time [76].

Bedbug infestations pose a global public health challenge, exacerbated by insecticide resistance and their ability to evade detection. Chemical insecticides are increasingly ineffective, necessitating alternative control methods, such as high-temperature washing (>60 °C), vacuuming, heat treatment, and freezing [82,83,84]. Future research should focus on potential vector–pathogen–host interactions and new disease transmission cycles influenced by globalization, climate change, and social inequalities.

## 5. Flea Bites and Associated Diseases

Flea bites are primarily caused by insects from the order *Siphonaptera*, which comprises 21 recognized families and approximately 2000 species of blood-feeding arthropods [85,86]. These insects range in size from 1.0 to 3.3 mm, featuring a flattened, thick exoskeleton with a glossy surface and highly developed limb musculature. Despite being wingless, fleas possess powerful hind legs, enabling them to jump nearly 200 times their body length [85,86].

Fleas are found worldwide, inhabiting trees, foliage, and shrubs, particularly in developed countries. From these environments, they leap onto vectors, including household pets—especially cats and dogs—where they prefer to settle around the ears, neck, back, and abdomen [87]. The primary flea species affecting humans is *Pulex irritans*, while the most common parasites of cats and dogs are *Ctenocephalides felis* and *Ctenocephalides canis* [87,88].

The flea life cycle consists of the following four stages: egg, larva, pupa, and adult (imago) [89]. Female fleas lay eggs on their hosts after each blood meal, with optimal temperature (15–35 °C) and humidity facilitating survival and development. Eggs hatch in 2 to 14 days, releasing larvae that feed on organic material over the next 18 days before weaving protective pupal cocoons using their own saliva. If environmental conditions are unfavorable, pupae can remain dormant for up to one year, awaiting optimal conditions before emerging as adults, typically in summer and humid environments [89].

Flea bites can cause various cutaneous hypersensitivity reactions, including the following:

Papular Urticaria: This condition is associated with type I (IgE-mediated) and type IV (cell-mediated) delayed hypersensitivity, resulting in pruritic, erythematous papular lesions. These lesions typically affect exposed skin areas, such as the ankles and feet and frequently follow a “breakfast, lunch, and dinner” pattern, characterized by clusters of three bites in a row [90]. Fleas are the leading cause of papular urticaria in developing regions of Africa, South America, and Asia, with a pediatric prevalence of 2.4–16.3% [90]. The characteristic lesion pattern is thought to result from flea saliva, which contains apyrase, an anticoagulant enzyme that triggers local hypersensitivity reactions [90].

Tungiasis: Caused by *Tunga penetrans*, the sand flea, this infestation is endemic to tropical and underdeveloped regions, particularly Africa, South America, and Central America [91,92,93]. The female *T. penetrans* burrows into soft skin areas, such as the subungual regions, toe webs, hands, and thighs, where she increases in size due to egg production, reaching 2–3 mm in diameter. This results in intradermal pressure, irritation, and local tissue damage, often leading to pain and discomfort. The initial lesion appears as a whitish papule with a central black crust, evolving into an erythematous, itchy nodule within a few days. In cases of secondary bacterial infection, the lesion may turn yellowish. Most infestations resolve spontaneously in 3–4 weeks following the death of the flea and natural skin exfoliation, but severe cases may develop into honeycomb-like lesions, leading to nail loss and walking difficulties [91,92].

Tungiasis should be suspected in travelers returning from endemic areas, particularly during August and September. Diagnosis is confirmed via biopsy, revealing flea exoskeleton, trachea, and eggs embedded in inflammatory tissue composed of eosinophils, lymphocytes, and plasma cells [93]. Treatment involves chloroform or ether application, followed by flea extraction with forceps. In complicated cases, surgical excision combined with systemic antibiotic therapy (e.g., metrifonate) may be necessary. Uncomplicated lesions heal with minimal scarring [93].

Fleas are vectors for multiple infectious diseases, including plague, cat scratch disease, and murine typhus [94,95,96].

Plague is a zoonosis caused by *Yersinia pestis*, a Gram-negative coccobacillus from the *Enterobacteriaceae* family [97]. The primary vector is the oriental rat flea (*Xenopsylla cheopis*); although, other hematophagous parasites, including lice and bedbugs, may also transmit the pathogen [97]. Plague remains endemic in parts of Africa, Asia, and South America, especially in rural, arid, and high-altitude regions [98,99,100]. Notably, it is also endemic, though less common, in the southwestern United States, where it persists in rodent populations, such as ground squirrels and prairie dogs, with fleas acting as vectors [101]. Globally, approximately 1000 to 2000 human cases are reported annually, with the majority occurring in Africa [102,103,104,105,106,107].

Transmission occurs when infected fleas regurgitate *Y. pestis* into the host during a blood meal [108]. Over 80 flea species are capable of carrying the bacterium, with vector efficiency varying by species. For instance, *Oropsylla tuberculata cynomuris* transmits the pathogen at 17.88% efficiency; whereas, *X. cheopis* transmits it at 6.4% efficiency [108].

Plague manifests in the following three clinical forms: bubonic, septicemic, and pneumonic. Following an incubation period of 3–6 days, initial nonspecific symptoms include high fever (40 °C), myalgia, arthralgia, headache, and chills [109,110,111,112,113].

Bubonic plague is the most common form, presenting with painful, swollen lymph nodes (buboes) at the femoral (31%), inguinal (24%), axillary (22%), or cervical (9%) regions [109,110,111,112,113]. Without early antibiotic treatment, the mortality rate reaches 60% but can be reduced to 5% with prompt therapy [109,110,111,112,113].

If untreated, *Y. pestis* can enter the bloodstream, leading to septicemic plague, which is characterized by hypotension, gastrointestinal symptoms, disseminated intravascular coagulation (DIC), purpura, and limb gangrene, with a high fatality rate within 24 h [114,115,116,117].

The pneumonic form is the deadliest, featuring pneumonia, cough, dyspnea, and hemoptysis. Mortality ranges from 25–50% with rapid diagnosis and treatment, reaching nearly 100% without intervention [114,115,116,117].

Diagnosis is based on bacterial culture, Gram-Giemsa or Wayson staining, PCR, and serology [114,115,116,117]. Rapid antigen tests for F1 capsular antigen are increasingly used [117].

According to WHO and CDC guidelines, first-line treatment for pediatric patients includes streptomycin or gentamicin, with doxycycline, chloramphenicol, or ciprofloxacin as alternatives for 10–14 days [118,119]. FDA-approved therapies for pneumonic or septicemic plague include ciprofloxacin and levofloxacin. Table 4 describes the treatment of plague. Exposed individuals require post-exposure prophylaxis with doxycycline or ciprofloxacin for seven days [118].

Cat scratch disease, caused by *Bartonella henselae*, is primarily transmitted through scratches or bites from infected cats but can also occur via flea bites, particularly *Ctenocephalides felis* [120,121]. While most cases result from direct cat contact (94.7%), 3.2% occur via flea transmission [122,123].

Pediatric patients under 18 years old are the most affected group (55%) [124,125]. After an incubation period of 7–12 days, patients develop regional lymphadenopathy, fever, headache, and fatigue. In 14% of cases, systemic symptoms include weight loss, hepatosplenic granulomas, and prolonged fever [126,127].

Diagnosis is confirmed by IgG serology, with elevated ESR and transaminases in systemic cases [120,121]. Treatment consists of azithromycin or trimethoprim–sulfamethoxazole, with rifampin for severe cases [126,127].

Murine typhus, caused by *Rickettsia typhi*, is transmitted via rat fleas (*Xenopsylla cheopis)*. The disease is endemic in Southeast Asia, Africa, California, and Texas [128]. Symptoms include fever, headache, rash, and interstitial pneumonia [129]. First-line treatment is doxycycline (4 mg/kg/day for ≥5 days), with tetracycline or chloramphenicol as alternatives [130].

## 6. Conclusions

Children are particularly susceptible to the health risks posed by arthropod bites, particularly from ticks, fleas, sand flies, and bedbugs, which serve as vectors for a range of infectious diseases. As climate change continues to alter environmental conditions, the geographic distribution of these vectors is expanding, increasing the incidence of vector-borne infections in previously unaffected regions. Rising temperatures, changes in humidity, and altered precipitation patterns may influence vector activity, breeding cycles, and disease transmission dynamics, making these infections an increasingly global public health concern.

Recent advancements in diagnostics, such as PCR testing and improved serologic assays, have enhanced the early and accurate identification of the pathogens responsible for these infections. However, diagnostic limitations persist, particularly in the early stages of infection or in regions with limited healthcare access, where delays in identification can lead to missed or misdiagnosed cases. The continued development of rapid, cost-effective, and widely available diagnostic tools is essential for timely disease detection and management.

Despite progress in medical treatments, prevention remains the most effective strategy for reducing the burden of these diseases. This requires a comprehensive, multidisciplinary approach that includes vector control measures, personal protective behaviors, and community education initiatives. Effective preventive strategies include avoiding high-risk areas, wearing protective clothing, and applying repellents, such as DEET, picaridin (icaridin), IR3535, and PMD (para-menthane-3,8-diol).

Parents and caregivers should be educated on the early recognition of symptoms that may indicate vector-borne diseases, enabling prompt diagnosis and intervention. Public health authorities must implement targeted vector surveillance and control programs, particularly in areas where vector populations are increasing due to climate and environmental changes. Surveillance systems should track emerging and re-emerging arthropod-borne diseases, allowing for early outbreak detection and response.

Vaccination remains an essential tool for disease prevention, particularly in endemic areas. TBE vaccine is a successful example of how immunization can significantly reduce disease incidence in high-risk regions. However, vaccines for most of these VBDs are still lacking, underscoring the need for continued research and investment in vaccine development. Future efforts should focus on expanding vaccine availability, affordability, and accessibility, particularly for low- and middle-income countries where these infections disproportionately occur.

The current review primarily organizes content by vector or pathogen type, lacking an integrative framework that emphasizes the unique vulnerabilities and combined risks faced by the pediatric population. This approach falls short of reinforcing the central claim that children should be prioritized in prevention strategies. Moreover, the review does not include any analysis of temporal trends or geographic distribution maps of these diseases among children in specific regions (e.g., the Mediterranean, South America, Southeast Asia), limiting the discussion on region-specific intervention strategies for high-risk pediatric groups. Incorporating such data would enhance the understanding of region-specific risks and inform targeted prevention efforts. In addition, while the review mentions vaccines and protective measures (e.g., DEET), it lacks detailed information on vaccines available or in development for children, their safety profiles, or age-appropriate dosage limits. The recommendations are, therefore, not sufficiently actionable, and the discussion does not consider parental knowledge, attitude, and practice (KAP) disparities or household-level risk factors. The review also does not address emerging arboviruses (e.g., Zika virus) or discuss pediatric-specific challenges during outbreaks (e.g., travel-associated risks, school-based transmission). Including this information would provide a more comprehensive understanding of the challenges and opportunities in protecting children from these diseases.

In conclusion, effectively combating arthropod-borne diseases requires a collaborative effort between scientific research, public health systems, and community engagement. Prioritizing prevention, early detection, and timely treatment is critical, particularly for children, who remain one of the most vulnerable populations. Addressing this growing health challenge necessitates ongoing research, strengthened global surveillance, and proactive public health interventions to mitigate the impact of vector-borne diseases in a rapidly changing world.

## Figures and Tables

**Table 1 pathogens-14-00553-t001:** Risk factors of tick-borne disease.

Risk Factors	Description
Living or Traveling in Endemic Areas	Areas where rickettsial diseases are common, including parts of North and South America, Africa, and Asia.
Outdoor Activities	Hiking, camping, or working in wooded areas or grasslands, where ticks and fleas are prevalent.
Occupational Risk	People working in agriculture, forestry, or veterinary fields are at higher risk due to frequent exposure to animals and insects.
Close Contact with Animals	Animals (especially rodents, cattle, and dogs) can be hosts for arthropods that carry rickettsial bacteria.
Poor Hygiene and Crowding	Environments with limited sanitation and crowded living conditions (e.g., prisons, shelters) can increase the risk of lice- and flea-borne rickettsiosis.
Climate	Warmer climates, particularly with seasonal changes (e.g., summer), support the growth and activity of vector organisms like ticks and fleas.
Immunocompromised States	Individuals with weakened immune systems (e.g., due to HIV, chemotherapy, or immunosuppressive drugs) may be more susceptible to severe disease.
Age	Both young children and the elderly may be at higher risk of severe complications from rickettsiosis.
Lack of Preventive Measures	Inadequate use of insect repellents, protective clothing, or tick checks can increase the risk of exposure.
Exposure to Arthropod Vectors	Rickettsiosis is transmitted through bites from infected ticks, fleas, or lice.

Adapted from references [1,15,16,17].

**Table 2 pathogens-14-00553-t002:** Clinical manifestations and treatment of major tick-borne diseases.

Disease	Common Symptoms	Treatment
Lyme disease	Rash (bullseye), fever, fatigue, headache, muscle and joint pain, lymphadenopathy, facial palsy, meningitidis and encephalopathy, atrio-ventricular block, syncope or dizziness.	Doxycycline, Amoxicillin or Cefuroxime Axetil if localized desease Ceftriaxone of Cefotaxime for disseminated desease
Rickettsiosis	Fever, headache, maculopapular rash, muscle aches, nausea, vomiting, loss of appetite	Doxycycline or Azithromycin
Tularemia	Fever, chills, myalgias, vomiting, fatigue, headache, skin ulcer at the site of infection, lymphadenopathy.	Doxycycline or Ciprofloxacin for mild forms Gentamicin and Streptomycin for severe forms
Babesiosis	Fever, chills, sweats, hemolytic anemia, splenomegaly, hepatomegaly, jaundice, fatigue, malaise, and disseminated intravascular coagulation (DIC)	Clindamycin and Quinine
Tick-borne encephalitis (TBE)	Fever, headache, stiff neck, vomiting, fatigue, meningeal signs, in severe cases: paralysis	No specific antiviral treatment, supportive care

Adapted from references [18,27].

**Table 3 pathogens-14-00553-t003:** Treatments for Chagas disease.

	Dosage ≥ 12 Years	Dosage < 12 Years	Healing Rate
Nifurtimox	Acute phase: 10–15 mg/kg in 3–4 doses for 60–90 days Chronic phase: 8–10 mg/kg for 60–90 days	Acute phase: 15 mg/kg in 3–4 doses for 60 days Chronic phase: 8–10 mg/kg for 60–90 days	86% in children and 7–8% in adults. Side effects in 43–97.5%: anorexia and weight loss, neurological disorders, nausea and vomiting, fever, and rash.
Benznidazole	Acute phase: 5–10 mg/kg in 2–3 doses for 60 days Chronic phase: 5–7.5 mg/kg for 60 days	Acute phase: 10 mg/kg in 2–3 doses for 60 days Chronic phase: 5–7.5 mg/kg for 60 days	Close to 100% in patients with congenital disease if treated within the first year of life, 76% in adults with acute disease, 60 to 93% in chronic children, and 2 to 40% in chronic adults. Most common side effects: rash, anorexia, asthenia, headache, and sleep disturbances.

**Table 4 pathogens-14-00553-t004:** Treatment of plague.

Antibiotic	Dose	Rout of Administration
Streptomycin	15 mg/kg twice daily (max 2 g/day)	IM
Gentamicin	2.5 mg/kg/dose every 8 h	IM or IV
Levofloxacin	8 mg/kg/dose every 12 h (max 250 mg/dose)	IV
Ciprofloxacin	15 mg/kg/dose every 12 h	
Doxycycline	Weight < 45 kg: 2.2 mg/kg twice a day (max 100 mg/dose) Weight > 45 kg 200 mg loading dose, then 100 mg every 12 h	IV
Chloramphenicol (if age > 2 years)	25 mg/kg every 6 h (maximum daily dose, 4 g)	IV

IM, intramuscular; IV, intravenous.

## Data Availability

Not applicable.

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
