# Peer review of "Beyond Mosquitoes: A Review of Pediatric Vector-Borne Diseases Excluding Malaria and Arboviral Infections"

_pathogens, 2025, doi:10.3390/pathogens14060553_

Round 1
Reviewer 1 Report
Comments and Suggestions for Authors
This manuscript reviews pediatric vector borne diseases. It is generally well written, and the areas included are well covered with only minor comments to be addressed.
The major issues that need to be addressed are two. First, "this review discusses the clinical manifestations, diagnostic challenges, and treatment options for major pediatric vector-borne diseases" but yet excludes the largest portion of pediatric VBDs, which are those involving disease agents transmitted by mosquitoes. The impact of malaria on children is huge. A review discussing the major pediatric VBDs should certainly include malaria and mosquito arboviral disease. If the intent of the Authors is to review VBDs other than those involving mosquito, this should be clearly delineated in the abstract and text. Along with this, details on whether this review followed standardized review scoping guidelines and methodologies is missing.
Second, in several places, good factual information is provided but poorly referenced. The introduction in particular lacks adequate citations.
Additional line comments below:
L35-56: These two paragraphs are well written, cohesive and they build a structure for what the reader can expect from the review. They both need to be referenced to support all the factual citation. Ref [1] is not the source of all the factual information in the first paragraph and similarly [2] is not the source of all the information in the second.
L61-62: Most reviews are currently conducted using a defined set of guidelines (like PRISMA) to ensure minimal bias and wide coverage of the relevant literature. What formal guidelines were used to conduct this review? Without additional detail on the methodology, a comprehensive electronic search may be adequate, or it may not. Mosquito VBD effects on children is completely missing.
L84: Is April to October correct or is it correct only for the Northern hemisphere? Note that some VBDs, like dengue, have different seasons between the Northern and Southern hemispheres. I do not know if this is true for tick borne infections.
Table 1: This summary needs citations.
Table 2: Needs citations.
L166 & L168: Phlebotomus specifically is the genus of "Old World" sandfly. Leishmania is transmitted in the Americas by sandlfy from the genus Lutzomyia. I think the better term to use is Phlebotominae as it encompasses both disease transmitting genera.
L206-207: This appears to be a nearly exact restatement of L175-177. Note they have different references as the source.
L222: Species name needs to be italicized.
L256: Cimicidae is not the correct taxonomic term that includes both groups covered in this section. The taxonomic divisions that include both are Cimicomorpha or Heteroptera.
L370: Italicize
L413: Plague also is endemic (but not common) in similar areas in Southwest North America in fleas on ground squirrels.
Author Response
This manuscript reviews pediatric vector borne diseases. It is generally well written, and the areas included are well covered with only minor comments to be addressed.
Re: Thank you for your comments. We revised our manuscript according to your suggestions.
The major issues that need to be addressed are two. First, "this review discusses the clinical manifestations, diagnostic challenges, and treatment options for major pediatric vector-borne diseases" but yet excludes the largest portion of pediatric VBDs, which are those involving disease agents transmitted by mosquitoes. The impact of malaria on children is huge. A review discussing the major pediatric VBDs should certainly include malaria and mosquito arboviral disease. If the intent of the Authors is to review VBDs other than those involving mosquito, this should be clearly delineated in the abstract and text. Along with this, details on whether this review followed standardized review scoping guidelines and methodologies is missing.
Re: Thank you for your valuable feedback. To address your concern and clarify the scope of our review, we have revised the title and abstract to explicitly state the exclusion of mosquito-borne diseases, including malaria and arboviral infections. This distinction ensures that readers understand our focus on other significant pediatric vector-borne diseases.
Second, in several places, good factual information is provided but poorly referenced. The introduction in particular lacks adequate citations.
Re: References have been added as requested.
Additional line comments below:
L35-56: These two paragraphs are well written, cohesive and they build a structure for what the reader can expect from the review. They both need to be referenced to support all the factual citation. Ref [1] is not the source of all the factual information in the first paragraph and similarly [2] is not the source of all the information in the second.
Re: Revised as suggested.
L61-62: Most reviews are currently conducted using a defined set of guidelines (like PRISMA) to ensure minimal bias and wide coverage of the relevant literature. What formal guidelines were used to conduct this review? Without additional detail on the methodology, a comprehensive electronic search may be adequate, or it may not. Mosquito VBD effects on children is completely missing.
Re: Thank you for your insightful feedback. To address your concerns and enhance the clarity and methodological rigor of our review, we have revised the methodology section accordingly. These revisions explicitly state the exclusion of mosquito-borne diseases, including malaria and arboviral infections, and provide a more detailed description of our search strategy and inclusion criteria.
L84: Is April to October correct or is it correct only for the Northern hemisphere? Note that some VBDs, like dengue, have different seasons between the Northern and Southern hemispheres. I do not know if this is true for tick borne infections.
Re: Revised.
Table 1: This summary needs citations.
Re: Added in the legend.
Table 2: Needs citations.
Re: Added in the legend.
L166 & L168: Phlebotomus specifically is the genus of "Old World" sandfly. Leishmania is transmitted in the Americas by sandlfy from the genus Lutzomyia. I think the better term to use is Phlebotominae as it encompasses both disease transmitting genera.
Re: Revised as suggested.
L206-207: This appears to be a nearly exact restatement of L175-177. Note they have different references as the source.
L222: Species name needs to be italicized.
Re: Revised.
L256: Cimicidae is not the correct taxonomic term that includes both groups covered in this section. The taxonomic divisions that include both are Cimicomorpha or Heteroptera.
L370: Italicize
Re: Revised.
L413: Plague also is endemic (but not common) in similar areas in Southwest North America in fleas on ground squirrels.
Re: Revised according to your comment.
Reviewer 2 Report
Comments and Suggestions for Authors
This review provides a comprehensive overview of the clinical manifestations, diagnostic challenges, and treatment options for common pediatric vector-borne diseases. It emphasizes the rising global transmission risk of these diseases due to climate change and increased human activity. The article notes that although effective treatments are available for some diseases, prevention remains the cornerstone of pediatric disease control, relying on multilayered strategies such as vector control, vaccination, and health education.
Although the review covers a wide range of vector-borne diseases, it organizes the content primarily by vector/pathogen type (tick, sandfly, bedbug, flea), lacking an integrative framework focused on the combined risks or specific susceptibilities of the pediatric population. As a result, the discussion falls short of reinforcing the central claim that children should be prioritized in prevention strategies.
The article does not include any analysis of temporal trends or geographic distribution maps of these diseases among children in specific regions (e.g., the Mediterranean, South America, Southeast Asia), which limits the discussion on region-specific intervention strategies for high-risk pediatric groups.
While it mentions vaccines and protective measures (e.g., DEET), the review lacks detailed information on vaccines available for children, their safety profiles, or age-appropriate dosage limits. The recommendations are therefore not sufficiently actionable, and the discussion does not consider parental KAP (knowledge, attitude, practice) disparities or household-level risk factors.
The review does not address emerging arboviruses (e.g., Zika virus), nor does it discuss pediatric-specific challenges during outbreaks (e.g., travel-associated risks, school-based transmission). Additionally, it fails to mention the current progress and barriers in pediatric vaccine development for vector-borne diseases.
Author Response
This review provides a comprehensive overview of the clinical manifestations, diagnostic challenges, and treatment options for common pediatric vector-borne diseases. It emphasizes the rising global transmission risk of these diseases due to climate change and increased human activity. The article notes that although effective treatments are available for some diseases, prevention remains the cornerstone of pediatric disease control, relying on multilayered strategies such as vector control, vaccination, and health education.
Re: Thank you for your comments. We revised the manuscript accordingly.
Although the review covers a wide range of vector-borne diseases, it organizes the content primarily by vector/pathogen type (tick, sandfly, bedbug, flea), lacking an integrative framework focused on the combined risks or specific susceptibilities of the pediatric population. As a result, the discussion falls short of reinforcing the central claim that children should be prioritized in prevention strategies.
Re: We considered this limitation in the Conclusions.
The article does not include any analysis of temporal trends or geographic distribution maps of these diseases among children in specific regions (e.g., the Mediterranean, South America, Southeast Asia), which limits the discussion on region-specific intervention strategies for high-risk pediatric groups.
Re: We considered this limitation in the Conclusions.
While it mentions vaccines and protective measures (e.g., DEET), the review lacks detailed information on vaccines available for children, their safety profiles, or age-appropriate dosage limits. The recommendations are therefore not sufficiently actionable, and the discussion does not consider parental KAP (knowledge, attitude, practice) disparities or household-level risk factors.
Re: We considered this limitation in the Conclusions.
The review does not address emerging arboviruses (e.g., Zika virus), nor does it discuss pediatric-specific challenges during outbreaks (e.g., travel-associated risks, school-based transmission). Additionally, it fails to mention the current progress and barriers in pediatric vaccine development for vector-borne diseases.
Re: We considered this limitation in the Conclusions.
Round 2
Reviewer 2 Report
Comments and Suggestions for Authors
none